# Soil Fungal Diversity of the Aguarongo Andean Forest (Ecuador)

**DOI:** 10.3390/biology10121289

**Published:** 2021-12-07

**Authors:** Ernesto F. Delgado, Adrián T. Valdez, Sergio A. Covarrubias, Solveig Tosi, Lidia Nicola

**Affiliations:** 1Laboratories Life Sciences, Research Group, Department of Environmental Engineering, INBIAM, Salesian Polytechnic University, Calle Vieja 12-30 and Elia Liut, Cuenca 010102, Ecuador; mdelgado@ups.edu.ec (E.F.D.); adrian.valdez@utalca.cl (A.T.V.); 2Academic Unit of Chemical Sciences, Campus Siglo XXI, University of Zacatecas, Carretera Zacatecas-Guadalajara km 6, La Escondida, Zacatecas 98160, Mexico; tiberio_claudio@hotmail.com; 3Mycology Laboratory, Department of Earth and Environmental Sciences, University of Pavia, Via S. Epifanio 14, 27100 Pavia, Italy

**Keywords:** Andes, environmental DNA, fungal biodiversity, metabarcoding, natural reserve, forest ecosystems tropical mycobiota vulnerable species

## Abstract

**Simple Summary:**

The Kingdom Fungi is one of the richest in species, most of which are still unknown. Many fungal species are hidden in the tropics, the area richest in biodiversity on earth. In this paper, a mycological analysis is presented on a vast number of soil samples collected in the Aguarongo forest, an important Andean Natural Reserve of Ecuador. The study was carried out by analyzing the total DNA extracted from the soil and unveiled a total of more than 400 species of fungi. The most abundant species belong to *Ascomycota* and *Mortierellomycota*; some are important beneficial fungi for the environments such as antagonistics of fungal pathogens or nematode predators, while others are well-known producers of nutraceutical and pharmaceutical compounds. Based on the results of this study, a picture of the mycodiversity of Aguarongo forest soil was obtained. This area hides a huge number of unknown fungal species that could be discovered; thus, the protection of the Aguarongo forest is mandatory.

**Abstract:**

Fungi represent an essential component of ecosystems, functioning as decomposers and biotrophs, and they are one of the most diverse groups of Eukarya. In the tropics, many species are unknown. In this work, high-throughput DNA sequencing was used to discover the biodiversity of soil fungi in the Aguarongo forest reserve, one of the richest biodiversity hotspots in Ecuador. The rDNA metabarcoding analysis revealed the presence of seven phyla: *Ascomycota*, *Basidiomycota*, *Mortierellomycota*, *Mucoromycota*, *Glomeromycota*, *Chytridiomycota*, and *Monoblepharomycota*. A total of 440 identified species were recorded. They mainly belonged to *Ascomycota* (263) and *Basidiomycota* (127). In *Mortierellomycota*, 12 species were recorded, among which *Podila verticillata* is extremely frequent and represents the dominant species in the entire mycobiota of Aguarongo. The present research provides the first account of the entire soil mycobiota in the Aguarongo forest, where many fungal species exist that have strong application potential in agriculture, bioremediation, chemical, and the food industry. The Aguarongo forest hides a huge number of unknown fungal species that could be assessed, and its protection is of the utmost importance.

## 1. Introduction

The Andes represent the largest mountain range on Earth and cross the western part of South America. In Ecuador, the inter-Andean valley is composed of humid areas at high altitudes, with physiognomies ranging from grasslands to forest formations, combined with andisol soils, which develop from volcanic ash and show little or moderate evolution, forming a unique ecoregion. The forest soils in the Andean highlands have high biodiversity and unique environmental characteristics. It is an endangered ecosystem, and little is known about its microbial community. Fungi represent an essential functional component of terrestrial ecosystems as decomposers and biotrophs (mutualists, pathogens, and necrotrophs), and they are one of the most diverse groups of Eukarya [1,2].

The soil fungal communities are affected by numerous biotic and abiotic factors, including seasons, soil characteristics, age and plant host species, and different soil managements [3,4]. The study of the ecological factors that underlie the dynamics of fungal communities remains a challenge due to its high taxonomic and ecological diversity.

Molecular methods based on PCR and ribosomal DNA sequencing have been successfully used to identify fungi at different taxonomic levels in different environmental samples and have helped to elucidate the ecological conditions that affect the structure and diversity of fungal communities [5,6,7]. The internal transcribed spacer region (ITS) is now widely used as a DNA barcode marker for the identification of fungal species [8]. Recent studies have demonstrated the potential of advances in DNA sequencing for quantifying and characterizing the fungal diversity, especially in those areas of the planet where biodiversity is very rich and complex [9].

Ecuador is one of the 17 megadiverse countries in the world, according to the World Conservation Monitoring Centre (WCMC) of the United Nations Environment Program. Ecuador has approximately 4.6 million hectares that have been declared as natural areas, which contain invaluable biodiversity and are a rich source of necessary ecological services. The fungal biodiversity of the natural areas in Ecuador is little known due to its extreme richness and complexity. The analysis of genetic material directly extracted from environmental samples coupled with DNA next-generation sequencing technology is useful in discovering its mycobiota and the monitoring of its biodiversity over time and in different ecological conditions [10].

Rich literature can be found on the fungal biodiversity of Ecuador, mainly focused on taxonomy and distribution of a single taxon at the species or hierarchical higher level [11,12,13,14,15]. Many papers focus on pathogenic fungi for important crops or on the relation between fungi and environments [16,17,18,19]. Recently, environmental fungal DNA was analyzed for community composition studies of disturbed areas [20,21] or specific substrates such as plants [22,23]. Investigation of mycobiota of natural areas in Ecuador focused on biodiversity, distribution, and ecology of mycorrhizae [24,25,26]. Data collected from environmental DNA concerning the entire mycobiota from natural areas and reserves of Ecuador are missing or rare.

In this work, high-throughput DNA sequencing was used to discover, for the first time, the biodiversity of soil fungi in the Aguarongo forest reserve, one of the richest biodiversity hotspots in the Ecuadorian Andes.

## 2. Materials and Methods

### 2.1. Declaration of Ethics

Specific permits were obtained for the field study described to collect 100 soil samples representative of the protected area. Samples did not include endangered or protected animal or plant species.

### 2.2. Sample Collection

The study site was located in Ecuador, in the Aguarongo forest, declared “protective vegetation area No. 10” by the Ecuadorian Ministry in 1985 (BVPA—El Bosque de Vegetación Protectora Aguarango) (Figure 1). The Aguarongo forest is one of the few fragments of Andean forest remaining in the Andean mountains in the province of Azuay and corresponds to low montane humid forest. The area, with an extension of 2080 ha, is the main water supply for the communities of the Gualaceo, Sigsig, and Cuenca cantons in southern Ecuador, resulting from its 191 streams [27,28]. It is located in the southern part of the middle basin of the Paute River, between 78°48′54″ and 78°52′22″ W, and 2°52′37″ and 2°59′43″ S. The altitude ranges between 2900 and 3320 m.a.s.l., and the average annual rainfall is 820 mm. The area is characterized by a dense closed forest, with 2–8 m tall tree cover, shrubs, and grass layer. The flora is dominated by *Weinmannia* spp. (sarar), *Eugenia* spp. (arrayan), *Podocarpus* spp. (huabisay), *Cerdela* spp. (Cedar), *Alnus acuminata* (alder), *Myrica pubescens* (laurel), and *Oreocallis grandiflora* (Gañal) [27].

The Aguarongo forest was divided into three altitudinal levels (L1, L2, L3; Table 1), and in November 2017, 35, 35, and 30 soil samples were taken for each level, making a total of 100 soil samples. Each sampling point was selected by a completely random design in an area of 10 m^2^. The number of samples per altitudinal level completely covered the extent of the altitudinal level. By means of a manual shovel, soil cores were taken (10 cm length × 10 cm width × 20 cm depth), and the shovel was disinfected with 75% ethanol after each sampling [29]. The soil samples were then placed in a cooler with dry ice for their conservation, transported to the laboratory, and stored at −20 °C (for metagenomic analyses) or 4 °C (for other analyses). Subsequently, animals, stones, and plant debris were removed, and the samples were sieved at 2 mm.

### 2.3. Soil Physicochemical Analysis

The soil physicochemical analyses were carried out on soil samples from each sampling site (L1, L2, L3) following the methods reported by Bloem et al. [30]. From each area (L1, L2, L3), three replicates were prepared by pooling 100 g aliquots derived from each soil sample. Each pooled sample was then sifted (2 mm mesh) and homogenized, based on the method reported by Uroz et al. [29].

Hydrogen potential was measured using an INESA pH meter (Shanghai REX Instrument Factory, Shanghai, China) in a 1:5 suspension of ultrapure water. The organic matter was evaluated following the Walkley–Black method [31]. The elements sodium, iron, zinc, manganese, copper, sulfur, calcium, magnesium, potassium, and phosphorus were measured using inductively coupled plasma–atomic emission spectrophotometry (ICP-AES), exchangeable aluminum and chlorides by ion chromatography, and nitrogen by Kjeldahl volumetric titration. Each measurement was performed in triplicate, and the mean for each sampling site was reported.

### 2.4. Environmental DNA Extraction from Soil Samples

Microbial DNA was extracted from 0.5 g of soil from each of the 100 samples using the Power Soil DNA isolation kit (MO BIO Laboratories, Carlsbad, CA, USA. U.U.), according to the manufacturer’s instructions. DNA extracted from five neighboring soil samples was combined, creating a total of 20 pooled samples (Appendix A), in order to optimize sequencing and to avoid pseudoreplication of sample points (the extension of the Aguarongo forest is limited). The quality and size of the DNA were verified by electrophoresis in 1% agarose gel. Additional quality control of the extracted DNA was performed, measuring 260/280 and 260/230 ratios. The DNA concentration was determined using Qubit Fluorometric Quantitation (Life Technologies, Carlsbad, CA, USA).

### 2.5. PCR Amplification and Next-Generation Sequencing (NGS, Illumina MiSeq)

Polymerase chain reaction (PCR) amplification of ITS hypervariable regions (6F-4R) was performed using the extracted DNA as a template for amplifying an internal fragment of the ITS gene. Primers ITS 3 (5′-GCA TCG ATG AAG AAC 132 GCA GC-3′), and ITS 4 (5′-TCC TCC TAT TGA TAT GC-3′), joined to a multiple identifier sequence (Illumina), were used [32]. For each sample, amplicons were generated in several duplicate PCRs using mixtures (25 μL) containing 25 pmol of each primer, 1x KAPA HiFi Hotstart Ready Mix (Kapa Biosystems, Wilmington, MA, USA), and 10 ng of the DNA template. The PCR program was performed following White et al. [32], consisting of an initial denaturation stage at 95 °C for 3 min, followed by 25 cycles at 95 °C for 30 s, 55 °C for 30 s, 72 °C for 30 s, and final extension at 72 °C for 5 min. Amplicons of the same treatment were grouped to reduce PCR variability and purified using Ampure accounts XP (Beckman Coulter, USA), according to the manufacturer’s instructions. After PCR cleaning, the Illumina sequencing adapters were joined by a second stage of PCR using the Nextera XT index kit (Illumina Inc., Sand Diego CA, EE U.U.). The mixture contained Nextera Index Primers 1 and 2 (5 µL), 2 KAPA HiFi HotStart ReadyMix (25 µL), DNA (5 µL), and PCR grade water (10 µL) for a total volume of 50 µL. The PCR program in this step consisted of an initial denaturation step at 95 °C for 3 min, followed by 8 cycles at 95 °C for 30 s, 55 °C for 30 s, 72 °C for 30 s, and a final step at 72 °C for 5 min. Amplicons were cleaned as described above. Amplicon libraries were quantified using Qubit (Invitrogen, CA, EE). The samples were combined in equimolar quantities (4 nM each) and were sequenced on a MisEq platform (paired-end sequencing 2 × 300, considering 2 × 50,000 reads/sample) of Illumina in Macrogen (Seoul, South Korea), according to the manufacturer’s instructions.

### 2.6. Taxonomic Allocation of Sequence Readings and Statistical Analysis

The paired-end reading sequences generated from Illumina MiSeq were processed using the software package “Quantitative Insights into Microbial Ecology 2” (QIIME 2, v2018.6) [33]. In short, the reads were trimmed, filtered, and merged with the DADA2 complement [34], keeping the sequences with a minimum quality score of 25, a minimum length of 240 bp for reverse readings, and a maximum length of 260 bp for advanced readings. Merged reads were collapsed into representative sequences or amplicon sequence variants (ASVs), then ASVs were filtered through de novo chimera using VSEARCH [35]. The sequences that were observed only once or twice (singletons and doubletons) were removed. The taxonomy of ASV was assigned at a 99% sequence identity based on the UNITE v7 database [36]. Nonfungal sequences were removed from the subsequent analysis, and the ASV table was rarefied to a uniform depth (100,000 sequences per sample) to reduce bias related to the depth of sequencing.

Taxonomy and shared files produced in QIIME were imported into R [37] using the Phyloseq package ver. 1.32.0 [38]. Alpha diversity was calculated using the Chao1, Simpson, and Shannon indices. Beta diversity was investigated using nonmetric multidimensional scaling (NMDS) on a Bray–Curtis distance matrix. The statistical test PERMANOVA was used, implemented in the vegan R package, ver. 2.5.6 [39], to assess any statistically significant difference among the fungal communities in the different sampling areas.

### 2.7. Access Numbers

High-performance sequencing datasets were deposited in the NCBI Biosamples data with access numbers SAMN11854455, SAMN11854456 197, and SAMN11854457 for Location 1, Location 2, and Location 3 ITS DNA metabarcoding libraries, respectively.

## 3. Results

### 3.1. Soil Chemical Characteristics

Concentrations of N, P, K, Mg, Ca, S, Cu, Mn, Zn, Fe, Na, exchangeable Al, and chlorides are presented in Table 2 for each site. The analyses revealed that some chemical parameters vary significantly among the sampling sites; altitudinal levels could be one of the conditions that influence the chemical properties of the soil, but type of soil must be considered as well.

The values of pH, N, P, K, Mg, Ca, S, Mn, and Cu are similar in L2 and L3 but are significantly different from L1. The content of Zn, Na, and exchangeable Al did not show differences between L1, L2, and L3. Site L1 showed the greatest differences, with lower values of N and P compared to the other locations, while in the case of Mg and Ca, it presents higher values.

### 3.2. Soil Fungal Assemblage Composition

The sequencing on the Illumina MiSeq Platform of the DNA of the 20 pooled soil samples produced a total of 3,699,594 reads, with an average of 184,979.7 reads per sample. After the cutting and filtering process, the reads were reduced to 2,794,683, with an average of 139,734 reads per sample. It was observed that L1 showed the highest abundance with an average of 833 OTUs, followed by L2 with an average of 780 OTUs and L3 with an average of 711 OTUs (Figure 2).

Metagenomic data were taxonomically organized in this work following the high-level classification of the Fungi reported by Tedersoo et al. [40]. The analysis revealed the presence of seven identified phyla in the soil samples: *Ascomycota* and *Basidiomycota* (subkingdom of *Dikarya*), *Mortierellomycota*, *Mucoromycota* and *Glomeromycota* (subkingdom *Mucoromyceta*), and *Chytridiomycota* and *Monoblepharomycota* (subkingdom *Chytridiomyceta*). The most abundant phylum was *Ascomycota*, with a relative abundance of 32–36%, followed by *Mortierellomycota* (25–28%), *Basidiomycota* (9–11%), and *Mucoromycota* (3–7%). The less abundant phyla were, as expected, *Glomeromycota*, (0.1%) *Chytridiomycota* (0.02%), and *Monoblepharomycota* (0.0005%). The identified phyla were equally distributed in each sampling site, except *Glomeromycota*, which was more abundant in Site L3, the highest in altitude. The share of fungal OTUs assigned to “Unidentified” was quite abundant in each site, reaching 17–35% of the total. In total, 408 taxa identified at the species level were found. All recorded species with their relative abundances in each sampling site are listed in Appendix A.

#### 3.2.1. Subkingdom Dikarya

##### *Ascomycota* 

Within the phylum *Ascomycota*, the analysis revealed the recording of 11 classes (Figure 3). The most abundant classes were *Sordariomycetes* (8–17% of the total fungi), mainly represented by the orders *Hypocreales* and *Chaetospheriales*; *Leotiomycetes* (3–9% of the total fungi), mainly represented by the order *Helotiales*; *Eurotiomycetes* (2–8% of the total fungi), mainly represented by *Eurotiales* and *Chaetotyriales*; and *Dothideomycetes* (2–5% of the total fungi) mainly represented by *Pleosporales* (Figure 4).

In *Ascomycota*, 263 identified genera were recorded; 58 genera reached at least 0.1% of the total fungi and were present in at least one of the samples with this value. Among them, the most abundant genera were: *Fusarium* (6–12%), *Penicillium* (0.5–6%), *Oidiodendron* (0.1–2%), *Bipolaris* (1–2%), and *Ilyonectria* (0.4–1%). The other genera were recorded at less than 0.01%.

The ascomycetous taxa identified at the species level were 235 (Appendix A). Most of them belonged to *Penicillium* (nine species); *Trichoderma* (six), *Oidiodendron* (five), *Cladonia*, *Cyphellophora*, *Exophiala*, *Peltigera*, and *Phialophora* (all with four species each), and *Acremonium, Aspergillus, Chalara, Chlonostachys, Hirsutella*, and *Metarhizium* (three species each).

##### Top Ten Most Abundant Species of *Ascomycota*

Among the identified taxa, *Fusarium oxysporum* s. lat. was, undoubtedly, the most abundant species, reaching 7–10% of the total fungal assemblage. *Pleotrichocladium opacum* was well represented (3%), together with *Penicillium spinulosum* and *Curvularia lunata* (2%). Other abundant identified species were *Trichoderma asperellum* and *Ilyonectria panacis* (1%), *Tetracladium apiense* and *Chaetosphaeria vermicularioides* (up to 0.5%), *Leptobacillium leptobactrum* (up to 0.4%), and *Penicillium donkii* (up to 0.3%).

##### *Basidiomycota* 

*Basidiomycota* was represented by eight classes (Figure 5); the most abundant ones were *Tremellomycetes*, represented by the order *Trichosporonales*, and *Agaricomycetes* (both, up to 11% of the total fungi), mainly represented by the orders *Agaricales* and *Thelephorales* (Figure 6).

The recorded genera of *Basidiomycota* were 118. Only 29 of these genera reached at least 0.1% of abundance of the total fungi. Among them, the most abundant ones were: *Cuphophyllus*, *Saitozyma*, and *Trichosporon* (up to 7% of the total fungi), *Tomentella* (up to 6%), and *Hygrocybe* (up to 2%).

The basidiomycetous taxa identified at the species level were 127 (Appendix A). Most of the identified species belonged to *Entoloma* (11 species), and *Lepiota* (5 species).

##### Top Ten Abundant Species of *Basidiomycota*

The following species were the most abundant: *Saitozyma podzolica* (3–4%), *Apiotrichum wieringae* (1–2.5%), and Tomentella testaceogilva (1%). *Apiotrichum dulcitum*, *Clitocybe nebularis*, *Fomes fomentarius*, *Porpolomopsis calyptriformis*, *Sarcodon atroviridis*, and *Serendipita vermifera* were represented by less than 0.2%.

#### 3.2.2. The Subkingdom *Mucoromyceta*

##### *Mortierellomycota* 

*Mortierellomycota* was the second highest represented phylum. In the class *Mortierellomycetes*, the analysis revealed the presence of 5 genera and 12 species (Appendix A). The most dominant species were *Podia verticillata* (50%), *Podia horticola* (21%), *Mortierella alpina*, and *Linnemannia gamsii* (12% each) (Figure 7).

##### *Mucoromycota* 

The three classes of *Mucoromycota* were all represented in the soil samples with a total of 19 species (Appendix A). *Mucoromycetes* (14 species) were dominated by *Mucor moelleri* (58%), *Mucor hiemalis*, *Sepmannia pineti* (both 11%), and *Mucor abundans* (10%) (Figure 8). *Umbelopsidiomycetes* (four species) was completely dominated by *Umbelopsis vinacea* (97%, Figure 8). *Endogonomycetes* was present with only one identified species, *Jimgerdemannia lactiflua*.

##### *Glomeromycota* 

The taxa belonging to this phylum were poorly identified at the species level. The detected species were seven (Appendix A), belonging to six genera: *Claroideoglomus claroideum* (2%) followed by *Acaulospora alpina, A. lacunosa, Racocetra castanea, R. fulgida, Archaeospora trappei*, and *Dentiscutata heterogama*.

##### The Top *Mucoromyceta*

Of the total fungi, *Podila verticillata* was strongly represented (12–14%), followed by *P. horticola* (5–6%), *Mortierella alpina* (2–6%), *Linnemannia gamsii* (3%), *Umbelopsis vinacea* (2.5–3%), *Mortierella biramosa* (0.5–1%), and *Mucor moelleri* (0.1–1%).

#### 3.2.3. The Subkingdom *Chytridiomyceta*

In *Chytridiomyceta*, the identified species were very rare (four, listed in Appendix A). The most abundant was *Rhizophlyctis rosea* (13%), belonging to *Chytridiomycota*, followed by *Spizellomyces plurigibbosus* (6%), *Olpidium brassicae* (1%), and *Gallinipes pseudodichotomus* (1%). In *Monoblepharomycota*, the only identified species was *Monoblepharella mexicana* (3%).

### 3.3. Soil Mycobiota Diversity in the Sampling Sites

The alpha diversity found in the fungal communities of the three locations (L1, L2, L3) was similar regarding the Shannon and Simpson Indices, while the Chao 1 Index, which focuses especially on the abundance of rare species, found a significant difference between locations L1 and L3 (324 ± 26 and 275 ± 29, respectively, *p* < 0.05 ANOVA, followed by Tukey’s pairwise test, Figure 9).

The analysis of beta diversity present in the samples did not show any significant difference in the structure of soil fungal communities in the three sampling locations (PERMANOVA, *p* > 0.05; Figure 10).

Venn’s analysis showed that 43% of all OTUs was shared between the three sampling sites. At Locations L1 and L2, the number of unique OTUs was similar (129 and 133, respectively), while L3 had a lower number of unique OTUs (97, corresponding to 10%) (Figure 11).

## 4. Discussion

The Aguarongo forest reserve is one of the most important biodiversity hotspots in Ecuador, and in recent years, it has been the subject of numerous research studies on flora, fauna, use of soils, and water balance [28,42].

As for all the tropics, Ecuador has huge biodiversity that has yet to be revealed, especially in natural reserves such as the Aguarongo forest. This is true above all for Fungi, a Kingdom often neglected despite its key role in the functionality of terrestrial ecosystems.

Regarding the use of next-generation sequencing in research of soil fungal communities, there are few studies in the tropics; however, a collection of metagenomic data is being implemented [43,44,45,46]. Most Ecuadorian studies on soil fungi describe the diversity of arbuscular mycorrhizal fungi (AMF) associated with crops in the Andean part of Loja and Quito [47,48]. There are also studies of the communities of AMF associated with plants growing in contaminated soils such as crude-oil-contaminated sites in the Amazon region of Ecuador [20]. Using modern molecular methods, it was possible to obtain a detailed account of soil fungal biodiversity of the Aguarongo forest and give the first list of soil fungi of this important reserve of biodiversity in Ecuador.

In this work, 408 fungal OTUs were identified at the species level. Together with *Ascomycota*, the most abundant phylum in our study was *Mortierellomycota*. Its high abundance was expected, as this result has been recorded in other studies on the mycobiota of soil in high altitude environments [44,46] or from extreme environments such as the Alps [49] or Antarctica [50,51,52]. Surprisingly, in the Aguarongo forest, the abundance of *Mortierellomycota* is similar to that of *Ascomycota*. All the OTUs referring to *Mucoromyceta* in this study were updated following the recent paper of Vandepol et al. [41], in which the phylogeny of *Mortierellaceae* was resolved through a synthesis of multigene phylogenetics and phylogenomics. The present work significantly improved the global geography and environmental records of *Mortierellaceae* and contributed to the knowledge on the ecology of these species, especially in an under-sampled region such as Ecuador. These data contribute to understanding the ecological function of *Mortierellaceae*, which remains mostly unknown. Species of *Mortierellaceae* are often isolated from soils, decaying leaves, and insects [53], due to their saprotrophic nature [54].

Regarding alpha diversity, it was similar among all sampling sites; the values reported here are similar to those found for other regions of the world with altitudes above 2000 m.a.s.l. [44].

In the whole soil mycobiota of Aguarongo, the most frequent species is a *Mortierellaceae* species, *Podila verticillate*, representing 12–14% of the total of identified OTUs. This species (reported with its basionym *Mortierella verticillata*) was first recorded in South America from mountainous environments (1950 m.a.s.l.) in Brazil [55]. Our findings expand the distribution of *P. verticillata* both in latitude and altitude. This is true for all of the species of *Mortierellaceae*, which were abundantly found in Aguarongo forest soil up to 3221 m.a.s.l.

Within the identified species of *Ascomycetes* in Aguarongo, important facultative plant pathogenic fungi were abundantly recorded, for example from the *Fusarium oxysporum* complex, a well-known group of taxa due to their many pathogenic forms and high frequency of isolation in soil [56], and *Curvularia lunata*, an important pathogen of maize [57]. In addition to the plant pathogens many saprotrophic fungi were recorded, such as *Pleotrichocladium opacum* and *Penicillium spinulosum*, a dominant taxon in the Aguarongo forest.

Beneficial fungi were not lacking, such as *Trichoderma asperellum*, an important biocontrol agent, applied in agriculture for controlling plant fungal pathogens and plant-growth promoters [58,59], as well as *Arthrobotrys musiformis Pochonia bulbillosa*, and *P. suchlasporia*, well-known nematophagous fungi [60,61]. Particularly important in the Aguarongo forest is the presence of the genus *Metarhizium*, a genus that shows wide biodiversity in this area, with species having high potential as biocontrol agents against insects, specifically *M. anisopliae*, *M. flavoviride*, and *M. lepidiotae*, [62]. Although their presence is occasional, many other species with importance because of their ecological role as entomopathogens or nematopathogens, and potential application for sustainable agriculture [63] are found in the Aguarongo forest. This includes *Beauveria brongniartii, B. bassiana, Dactylella mammillata, Hirsutella minnesotensis, H. rhossiliensis, H. vermicola, Paecilomyces farinosus*, and *P. marquandii*.

*Basidiomycetes* were dominated by *Saitozyma podzolica*, a common yeast and strong decomposer of dead plant biomass isolated from soils worldwide [64]. *Apiotrichum wieringae* was also widely present in the soil of Aguarongo, a nonpathogenic member of the *Tricosporonaceae* family, able to degrade uric acid and aromatic compounds [65]. *Tomentella testaceogilva*, abundant in Aguarongo, is reported as a terricolous or lignicolous fungus associated with *Alnus*, but also with moss [66]. Although rarely recorded in the Aguarongo forest, a famous member of *Basidiomycetes* must be highlighted, namely *Fomes fomentarius*, an important plant pathogenic fungus rich in pharmacological compounds [67].

Fungi in soils of Aguarongo include members of the phylum *Glomeromycota*, which remain mainly unidentified. Some identified species are important arbuscular mycorrhizal fungi; the most abundant one was *Claroideoglomus claroideum*, considered a good plant promoter and heavy metal decontaminant [68]. In the Aguarongo forest, *Chytridiomycota* were dominated by *Rhizophlyctis rosea*, a highly effective plant biomass degrader and zoosporic fungus, commonly observed near the soil surface. This species has light-sensitive proteins that allow this fungus to remain in the euphotic zone [69]. Within *Monoblepharomycota*, only one species was recorded: *Monoblepharella mexicana*. The southernmost record of this species was reported by Steciow and Arambarri [70], while the present finding can be considered the record of this species at the highest elevation.

The analysis reported in this work is based on a large number of soil samples (100), thus giving a good representation of the fungal biodiversity of Aguarongo forest soil. The collection of samples along an altitudinal gradient did not reveal significant differences among the structure of the soil fungal communities in the three sampling locations (L1, L2, L3). Only Location L3 has a lower abundance of rare taxa with respect to L1, consistent with the Venn diagram.

The analysis of the records, reported in Appendix A, shows the Aguarongo forest soils as a natural area rich in fungal biodiversity, with the main phyla represented by a large number of taxa, known to be beneficial fungi. In addition, the Aguarongo forest is shown to be a rich source of species having strong application potential in agriculture and in the chemical and food industry. This area hides a huge number of unknown species that could be assessed, and its protection is of the utmost importance.

## Figures and Tables

**Figure 1 biology-10-01289-f001:**
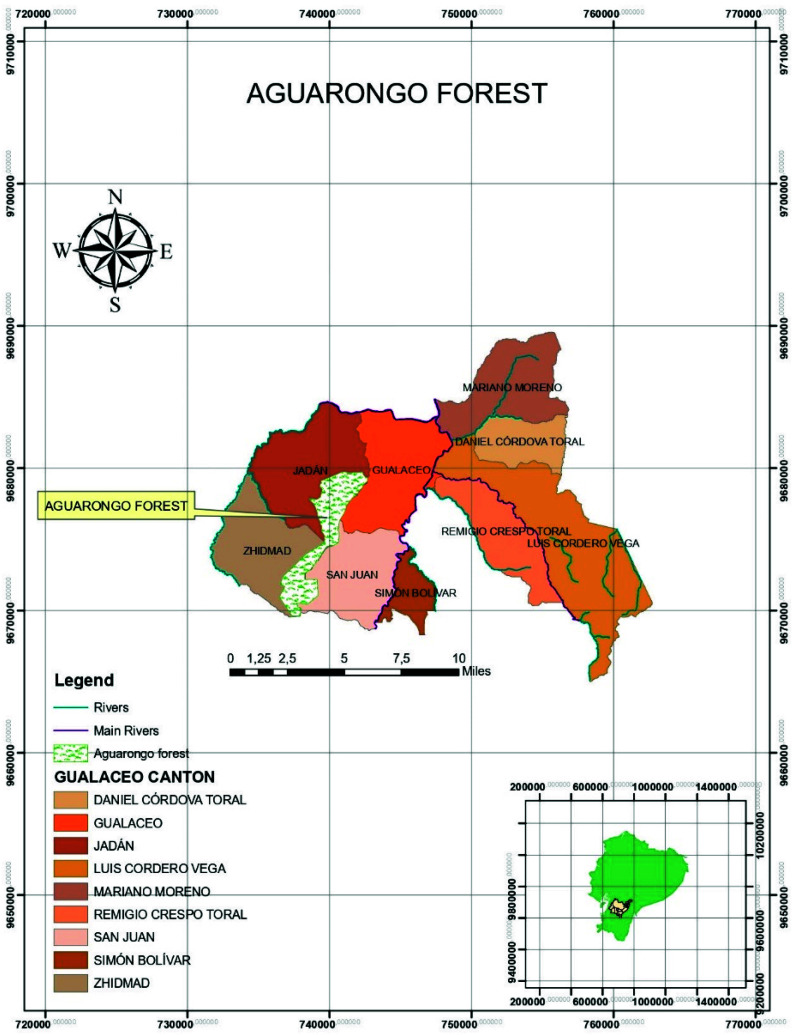
The upper part shows the location of the Aguarongo protective forest within the area that corresponds to the Gualaceo canton and its parishes, province of Azuay, Ecuador. The lower part of the figure shows the map of Ecuador with the location of the province of Azuay.

**Figure 2 biology-10-01289-f002:**
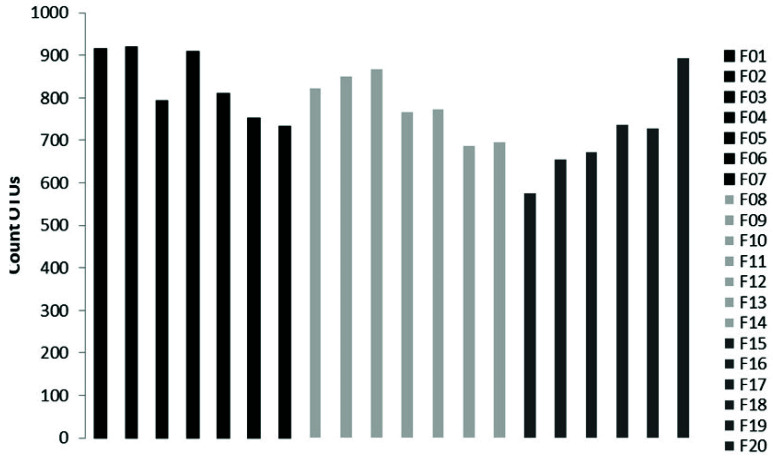
Number of OTUs present in each sample, with F1, F2, F3, F4, F5, F6, F7 belonging to altitudinal level L1, F8, F9, F10, F11, F12, F13, F14 beloning to L2, and F15, F16, F17, F18, F19, F20 belonging to L3.

**Figure 3 biology-10-01289-f003:**
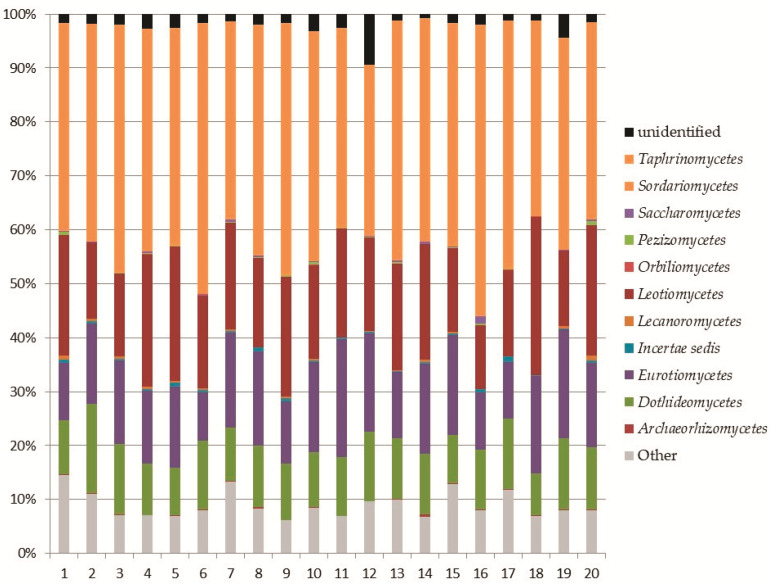
Relative abundance of *Ascomycota* classes in the soil samples.

**Figure 4 biology-10-01289-f004:**
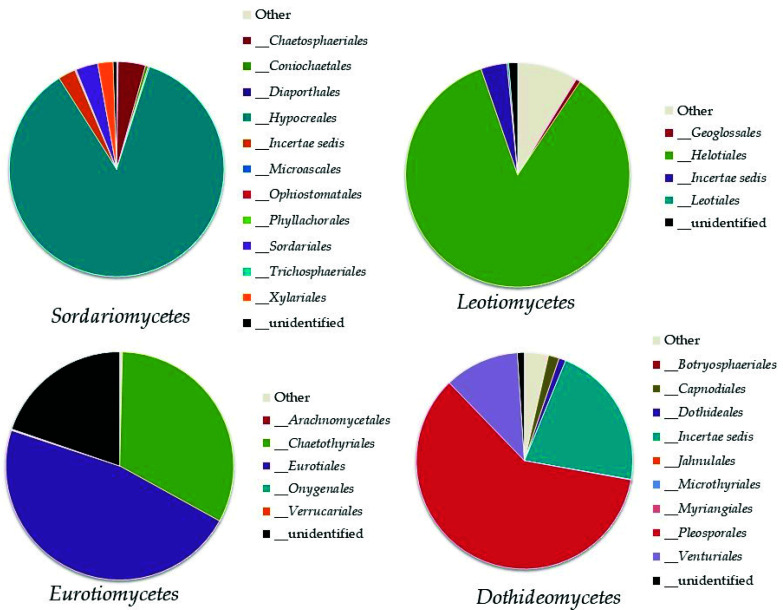
Order abundance in the most highly represented classes of *Ascomycota*: *Sordariomycetes*, *Leotiomycetes*, *Eurotiomycetes*, and *Dothideomycetes*.

**Figure 5 biology-10-01289-f005:**
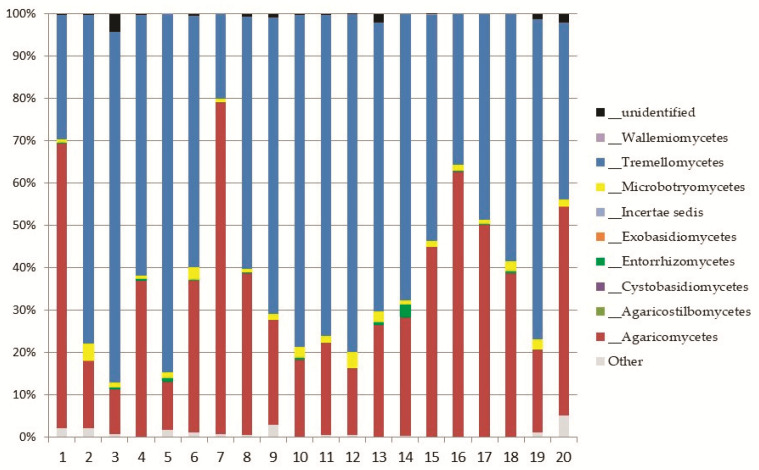
Relative abundance of classes of *Basidiomycota* in the soil samples.

**Figure 6 biology-10-01289-f006:**
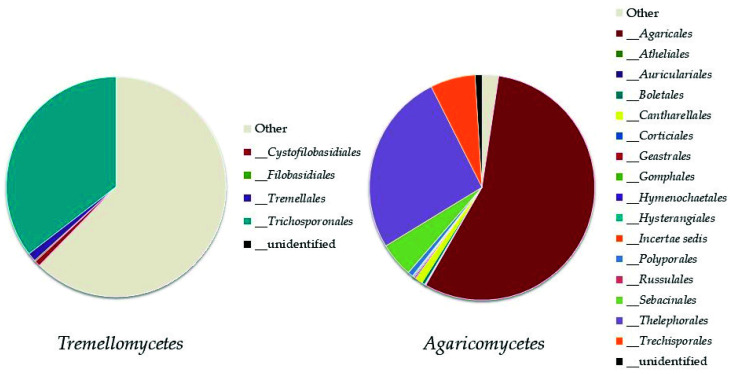
Abundance of orders in the most represented classes of *Basidiomycota: Tremellomycetes* and *Agaricomycetes*.

**Figure 7 biology-10-01289-f007:**
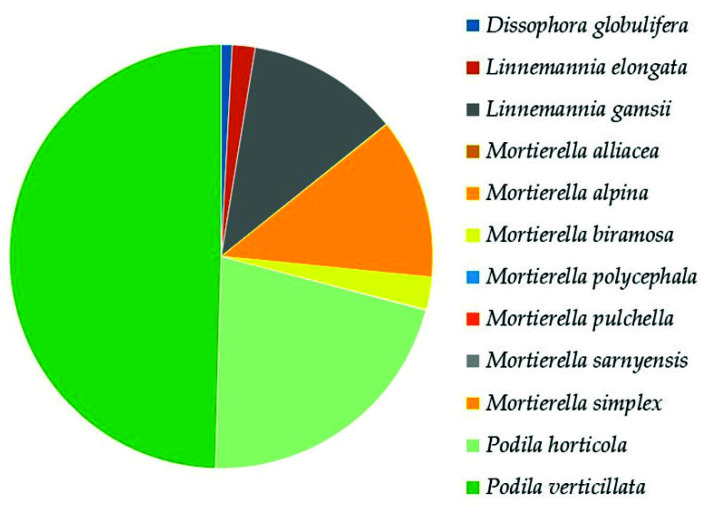
Relative abundance of species in *Mortierellomycetes* (*Mortierellomycota*), following the most recent taxonomic classification at the species level [41].

**Figure 8 biology-10-01289-f008:**
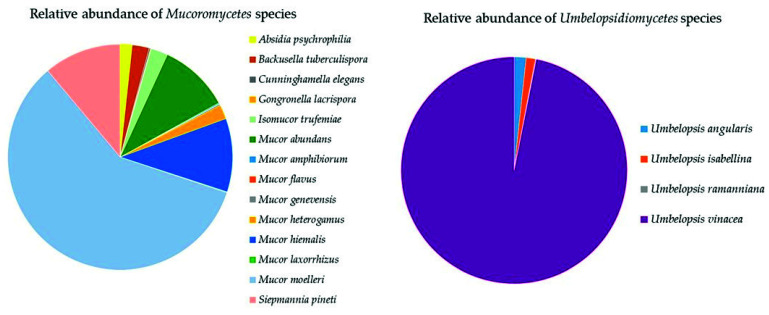
Relative abundance of species of *Mucoromycota* in the classes *Mucoromycetes* and *Umbelopsidiomycetes*.

**Figure 9 biology-10-01289-f009:**
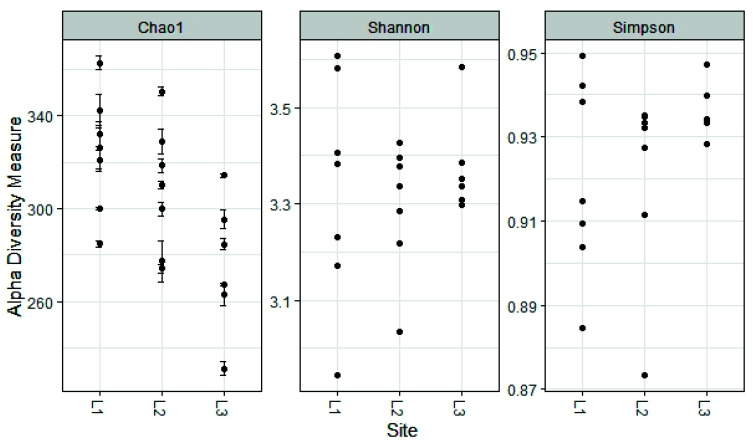
Estimated alpha diversity for different sampling sites.

**Figure 10 biology-10-01289-f010:**
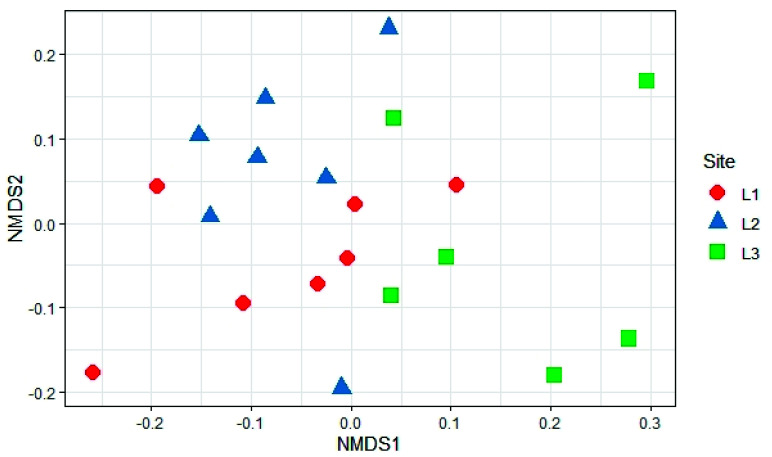
Non-Metric Multidimensional Scaling (NMDS) based on Bray-Curtis distances of soil fungal communities in the different sample locations (L1, L2, and L3) in the Aguarongo forest.

**Figure 11 biology-10-01289-f011:**
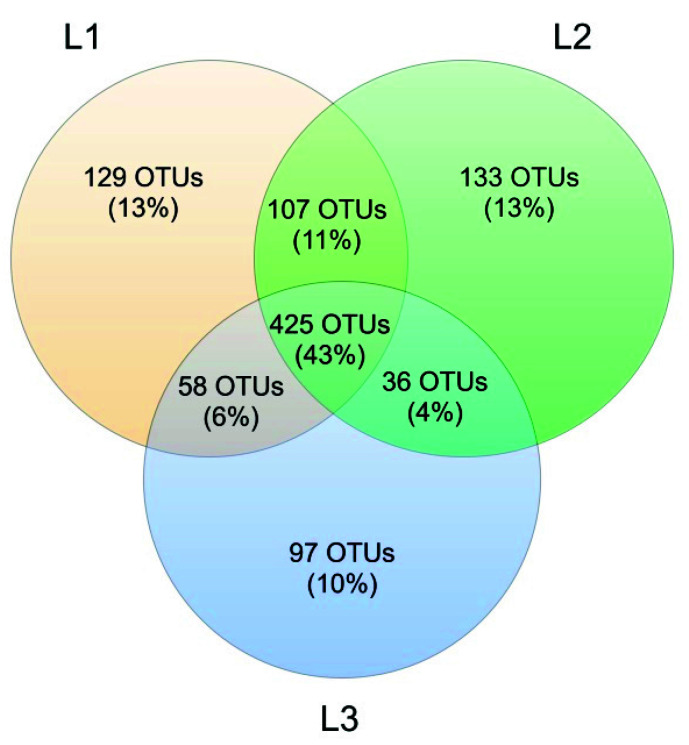
Venn diagram showing OTUs shared among the 3 sampling sites (L1, L2, and L3).

**Table 1 biology-10-01289-t001:** Sampling sites and number of soil samples.

Sampling Site	Number of Collected Samples	Altitude	Coordinates
L1	35	3101–3146	2.936° S; 78.8339° W
L2	35	3159–3173	2.936° S; 78.842° W
L3	30	3175–3221	2.959° S; 78.855° W

**Table 2 biology-10-01289-t002:** Soil chemical properties. Values are expressed in mg/kg for each compound, except Al expressed in meq/100 g, while N and organic matter are expressed in % s.m.s. Values with different letters in the same column are significantly different (*p* < 0.05).

**Sampling Sites**	**pH**	**Organic Matter**	**N**	**P**	**K**	**Mg**	**Ca**	**S**
L1	4.9 ^b^	10.9 ^a^	0.68 ^b^	3.6 ^b^	203.3 ^b^	425.7 ^b^	1776.3 ^b^	846.4 ^b^
L2	4.8 ^a^	15.5 ^a^	0.82 ^a^	5.9 ^a^	128.0 ^a^	162.7 ^a^	564.7 ^a^	1012.6 ^a^
L3	4.6 ^a^	12.3 ^a^	0.71 ^a^	5.3 ^a^	126.7 ^a^	177.7 ^a^	885.3 ^a^	790.4 ^a^
**Sampling Sites**	**Cu**	**Mn**	**Zn**	**Fe**	**Na**	**Cl**	**Al**	
L1	10.5 ^b^	118.7 ^b^	16.0 ^a^	834.3 ^a^	28.7 ^a^	11.8 ^a^	6.4 ^a^	
L2	8.7 ^a^	72.7 ^a^	15.7 ^a^	926.0 ^a^	31.3 ^a^	15.1 ^a^	6.9 ^a^	
L3	12.7 ^a^	107.7 ^a^	16.0 ^a^	1197.3 ^b^	24.3 ^a^	371.8 ^b^	7.1 ^a^	

## Data Availability

High-performance sequencing datasets were deposited in the NCBI Biosamples data with access numbers SAMN11854455, SAMN11854456 197, and SAMN11854457 for Location 1, Location 2, and Location 3 ITS DNA metabarcoding libraries, respectively.

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
