# Peer review of "Soil Fungal Diversity of the Aguarongo Andean Forest (Ecuador)"

_biology, 2021, doi:10.3390/biology10121289_

Round 1

Reviewer 1 Report

Here is the review of the paper entitled "Soil Fungal Diversity of the Aguarongo Andean Forest (Ecuador)" written by Ernesto F. Delgado and his co-authors.

The aim of the study was to assess the composition of soil fungal communities in Aguarongo Forest Reserve (Ecuador) which is one of the biodiversity hotspots in Ecuador. Assessment of fungal communities was based on metabarcoding of soil samples taken on three sampling subsites differing in altitude. The analysis of high-throughput sequencing data (Illumina) based on ITS sequence revealed the presence of 7 fungal phyla: Ascomycota, Basidiomycota, Mortierellomycota, Mucoromycota, Glomeromycota, Chytridiomycota and Monoblepharomycota. The analysis yielded 440 identified species in the whole studied area. The most common fungal phyla were Ascomycota (263 sp.) and Basidiomycota (127 sp.).

The study is interesting since there is a lack of fungal diversity studies in the tropical areas of the world that are threatened due to human influence and climate changes. But the study has some major issues that should be properly addressed before possible publication. The methods used are not sufficiently explained (see my comments in the pdf attached). Also, the record of "Porpolomopsis calyptriformis" is not well interpreted because from the literature records there are at least two taxa (one European and one North/South American) in the complex (see my comments). Also, some sentences are not easily understandable, so those should be rephrased and written more clearly. Please, go through my comments in the pdf attached. The English language requires moderate changes.

After the major revision, the paper could be reconsidered for publication in Biology journal.

Best, Reviewer

Author Response

Response to Reviewer 1 Comments

Point 1: The methods used are not sufficiently explained (see my comments in the pdf attached)

Response 1: We added additional information about the methods (lines 118-128, 156-159). We also added, as Supplementary Table 1, information on how the samples were pooled together.

Point 2: Also, the record of "Porpolomopsis calyptriformis" is not well interpreted because from the literature records there are at least two taxa (one European and one North/South American) in the complex (see my comments).

Response 2: We removed every mention to red lists in association to Porpolomopsis calyptriformis.

Point 3: some sentences are not easily understandable, so those should be rephrased and written more clearly. Please, go through my comments in the pdf attached. The English language requires moderate changes.

Response 3: we changed the manuscript according to you suggestions.

Please, see the attached pdf documents with all our responses to you comments on the manuscript.

Reviewer 2 Report

THE Manuscript presents important information on fungal diversity from Ecuador. Minor details must be checked.

Please, check for Chytridiomyceta were dominated by...

line122- correct the names: Steciow and Arabarri [72], 

Reviewer 3 Report

This paper provides an exciting analysis of the soil fungi in a South American natural area in which a number of beneficial and endangered fungi were discovered. Using the ITS sequence to analyze the DNA, more than 400 species of fungi could be identified to some level. Many editorial changes including misspellings are suggested on the manuscript. A few important changes are made such as the synonymy of Hypocrea with Trichoderma. Towards the end of the discussion I was not sure what the authors meant by stating that this was the uppermost record for Monoblepharella mexicana. Do you mean the highest elevation? Not most northern because it was described from Mexico.

In briefly checking the names of the fungi in the supplementary table, I find a number that need to be changed:

Gibberella baccata is now Fusarium lateritium

Haematonectria haematoccoca is now Neocosmospora haematococca

Hypocrea patella is now Trichoderma patella

Hypocrea sinuosum is now Trichoderma sinuosum

Hypocrea silvae-virgineae is now Trichoderma silvae-virgineae

Lalaria inostiophila is now Taphroina inositophila

Nectria balsamea is now Pleonectria balsamea

Nectria mariannaeae is now Mariannaea pinicola

I’m not familiar with the Basiodiomycota and hopefully there are not so many names changes in that group.

Author Response

Response to Reviewer 3 Comments

Point 1: Many editorial changes including misspellings are suggested on the manuscript.

Response 1: We changed the manuscript according to your suggestions.

Point 2: A few important changes are made such as the synonymy of Hypocrea with Trichoderma.

Response 2: We changed the names of Hyprocrea spp. to Trichoderma as you suggested.

Point 3: Towards the end of the discussion I was not sure what the authors meant by stating that this was the uppermost record for Monoblepharella mexicana. Do you mean the highest elevation? Not most northern because it was described from Mexico.

Response 3: Yes, we meant that it was the record of Monoblepharella mexicana at highest elevation, we corrected it in the text (Lines 422-423)

Point 4: In briefly checking the names of the fungi in the supplementary table, I find a number that need to be changed:

Gibberella baccata is now Fusarium lateritium

Haematonectria haematoccoca is now Neocosmospora haematococca

Hypocrea patella is now Trichoderma patella

Hypocrea sinuosum is now Trichoderma sinuosum

Hypocrea silvae-virgineae is now Trichoderma silvae-virgineae

Lalaria inostiophila is now Taphroina inositophila

Nectria balsamea is now Pleonectria balsamea

Nectria mariannaeae is now Mariannaea pinicola

I’m not familiar with the Basiodiomycota and hopefully there are not so many names changes in that group.

Response 4: We changed all the indicated taxa and we checked also the other phyla and changed to the latest name version. We changed the manuscript accordingly (lines 276, 308-309, 313, 340).

Round 2

Reviewer 1 Report

Second round review

The paper is modified according to my suggestions, although there are a few additional improvements required now:

  1. Line 105:  (33, 33 and 34 soil samples) is not in accordance with table 1. (35, 35, 30 samples). Please check!
  2. Line 273: of classes of Basidiomycota classes -> of classes of Basidiomycota
  3. Line 308: This phylum was poorly represented to the species level -> The taxa belonging to this phylum were poorly identified at the species level.
  4. Line 411: Basidiomicetes -> Basidiomycetes
  5. Lines 438-440: Please check names of authors since Sergio A. Covarrubias should be included now instead of Hernández S.

After abovementioned improvements the paper can be accepted for publication in Biology journal.

Best, Reviewer

Author Response

Responses to Reviewer 1 comments

1. Line 105:  (33, 33 and 34 soil samples) is not in accordance with table 1. (35, 35, 30 samples). Please check

Response 1: The number of samples in line 105 has been corrected and now it matches supplementary table 1

2. Line 273: of classes of Basidiomycota classes -> of classes of Basidiomycota

Response 2: Corrected (line 273)

3. Line 308: This phylum was poorly represented to the species level -> The taxa belonging to this phylum were poorly identified at the species level.

Response 3: Corrected (lines 308-309)

4. Line 411: Basidiomicetes -> Basidiomycetes

Response 4: Corrected (line 412)

5. Lines 438-440: Please check names of authors since Sergio A. Covarrubias should be included now instead of Hernández S.

Response 5: We substituted Covarrubias S.A. to Hernandez S in lines 439-441.